# scGRN-Entropy: Inferring cell differentiation trajectories using single-cell data and gene regulation network-based transfer entropy

Rui Sun[1,2], Wenjie Cao[3], ShengXuan Li[1,2], Jian Jiang[1,2], Yazhou Shi [1,2]*, Bengong Zhang [1,2]*

**1** School of Mathematical & Physical Sciences, Wuhan Textile University, Wuhan, Hubei, China, **2** Center for Applied Mathematics and Interdisciplinary Studies, Wuhan Textile University, Wuhan, Hubei, China, **3** School of Mathematics, Sun Yat-sen University, Guangzhou, Guangdong, China

* yzshi@wtu.edu.cn (YS); benyan1219@126.com (BZ)

**Data Availability Statement:** All datasets used in the paper and supporting information can be downloaded at https://zenodo.org/records/1443566 The code used in the paper and

## Abstract

Research on cell differentiation facilitates a deeper understanding of the fundamental processes of life, elucidates the intrinsic mechanisms underlying diseases such as cancer, and advances the development of therapeutics and precision medicine. Existing methods for inferring cell differentiation trajectories from single-cell RNA sequencing (scRNA-seq) data primarily rely on static gene expression data to measure distances between cells and subsequently infer pseudotime trajectories. In this work, we introduce a novel method, scGRN-Entropy, for inferring cell differentiation trajectories and pseudotime from scRNA-seq data. Unlike existing approaches, scGRN-Entropy improves inference accuracy by incorporating dynamic changes in gene regulatory networks (GRN). In scGRN-Entropy, an undirected graph representing state transitions between cells is constructed by integrating both static relationships in gene expression space and dynamic relationships in the GRN space. The edges of the undirected graph are then refined using pseudotime inferred based on cell entropy in the GRN space. Finally, the Minimum Spanning Tree (MST) algorithm is applied to derive the cell differentiation trajectory. We validate the accuracy of scGRN-Entropy on eight different real scRNA-seq datasets, demonstrating its superior performance in inferring cell differentiation trajectories through comparative analysis with existing state-of-the-art methods.

## Author summary

It is very important to study cell differentiation because it can help us understand the fundamental processes of life, elucidates the intrinsic mechanisms underlying diseases such as cancer, and advances the development of therapeutics and precision medicine. However, the existed methods for this usually much more rely on static gene expression data. They ignore the dynamical behavior of it. In this paper, we introduce method named scGRN-Entropy for inferring cell differentiation trajectories and pseudotime from scRNA-seq data. Our method divides cellular differentiation relationships into static and

supporting information can be download at: https://github.com/SuunRui/scGRN-Entropy.

**Funding:** This work was supported by the National Natural Science Foundation of China (NSFC) 11971367 to BGZ, 12371500 to BGZ). The funders had no role in study design, data collection and analysis, decision to publish, or preparation of the manuscript.

**Competing interests:** The authors have declared that no competing interests exist.

dynamic types. Static relationships are calculated based on gene expression levels, while dynamic relationships are derived from the similarity of cellular GRNs. We obtain the GRN from ordinary differential equations of gene expression, reflecting the internal dynamic regulatory relationships within cells. Incorporating GRNs into trajectory inference considers both biological reality and real datasets, it shows that our method can infer the cell differentiation trajectories much more accurately.

## Introduction

Cell differentiation is a pivotal process in the development and functioning of multicellular organisms, driving the specialization of cells to perform distinct functions [1]. Understanding the mechanisms underlying cell differentiation is essential for deciphering the complexities of biological development, maintaining homeostasis, and identifying pathological alterations that lead to disease [2, 3]. Recent advancements in single-cell RNA sequencing (scRNA-seq) have revolutionized the study of cell differentiation by enabling the high-resolution profiling of gene expression at the single-cell level [4–6]. For example, this technology allows researchers to capture the heterogeneity within cell populations, map dynamic changes during differentiation, identify specific gene expression patterns and key regulatory pathways, and construct detailed cellular trajectories that offer insights into the temporal progression of differentiation and its molecular determinants [7, 8]. The ability to infer cell trajectories from scRNA-seq data not only enhances our understanding of developmental biology but also has significant implications for regenerative medicine, disease modeling, and therapeutic interventions [9–13].

The current research on the process of cell differentiation mainly involved two differentiation information: the reconstruction of cell differentiation trajectory and the pseudotime ordering of cell differentiation [14]. The reconstruction of cell differentiation trajectory focused on the "differentiation history" of different types of cells, exploring the lineage relationship between different types of cells, and the study of cell differentiation process was more like a discrete description [15]. The pseudotime sequencing of cell differentiation focused on the stages of different cells in the process of differentiation, exploring the sequence of cell states, and its research on the process of cell differentiation was more similar to a continuous characterization [16]. Since Monocle [17] was proposed in 2014, many scholars had made important contributions to these two aspects. Wanderlust extracted continuous trajectories from snapshots of the system, rather than from time series data, and required only an approximate starting point as prior information [18]. Then, their team continued to develop Monocle2 [19], using the MST algorithm to learn cell trajectories and updating cell positions by moving cells to the nearest vertex in the MST. Monocle2 repeats this process until the cell trajectories and positions stabilize. Finally, the pseudotime of cells is calculated by the geodesic distance from the root vertex to the MST. Slingshot consists of two main stages: global lineage structure inference and pseudotime inference of cells along each lineage. In the first stage, Slingshot uses a cluster-based MST to robustly identify the key elements of the global lineage structure. For the second stage, Slingshot introduces a new method called simultaneous principal curves to fit smooth branching curves of these lineages, thereby translating the knowledge of the global lineage structure into stable estimates of cell-level pseudotime variables for each lineage [20]. Laleh Haghverdi et al. proposed the DPT algorithm, which was a random walk-based distance that computs pseudotime based on the simple Euclidean distance in the "diffusion map space" [21]. Caleb Weinreb et al. proposed the PBA algorithm, which was based on the population balance equation to calculate the pseudotime during cell differentiation

according to the first arrival time to the terminal cell [22]. Na Sun et al. proposed the scEpath algorithm, which used cell population data to generate a computational model referring to "biological differentiation time" and applied it to single-cell data to calculate the pseudotime of cell differentiation by unbiased association of cell cycle checkpoints with the internal molecular timer of a single cell [23]. Gioele La Manno et al. proposed the concept of RNA velocity [24], which is a high-dimensional vector that can predict the future state of individual cells on a timescale of hours. They measured the state transition relationships between cells by constructing ordinary differential equations for the changes in the amounts of spliced and unspliced mRNA, thereby constructing a KNN graph of cell differentiation to visualize the differentiation relationships of cell lineages. Subsequently, many researchers have conducted extensive studies in the field of RNA velocity, including UniTVelo [25], cellDancer [26], TFvelo [27], and scVelo [28], among others. In addition to the above algorithms, there were also many algorithms used for trajectory inference and pseudotime calculation of cell differentiation, such as scFates [29], SpaceFlow [30] and DBCTI [31]. Most algorithms calculate pseudotime based on the positional relationship between all cells and root cells in the cell differentiation trajectory or through the positional relationship in the gene expression space to infer the trajectory. However, the positional relationship of cells in the gene expression space reflects only their static state at a particular moment and does not capture their dynamic changes. Consequently, the calculation of pseudotime becomes dependent on the inferred trajectory, limiting the accuracy of these methods.

In this paper, we propose scGRN-Entropy, a novel method for inferring pseudotime and trajectory of cell differentiation based on scRNA-seq data by combining the static positional relationships of cells in gene expression space with the dynamic changes in cell states. To capture these dynamic changes, we introduce a method for resolving dynamic GRNs from scRNA-seq data. Based on the coupling of static and dynamic positions of cells, we construct an undirected graph of k-nearest neighbor cells. Using the dynamic features of cells obtained from GRN space, we infer the pseudotime trajectory of cells based on cell entropy. Finally, the differentiation relationships among cells in the lineage are established using the Minimum Spanning Tree (MST). Pseudotime is then further adjusted based on these differentiation relationships, thereby achieving a continuous pseudotime representation of cell differentiation.

## Materials and methods

### Constructing GRN for each cell

To construct the gene regulatory network for each cell (Fig 1A (II)), we first perform gene pooling [32], which involves dimensionality reduction by clustering genes based on their correlations across all cells. That is, for genes $x$ and $y$, the Pearson correlation coefficient $r_{xy}$ can be calculated by

$$r_{xy} = \frac{\sum (x_i - \bar{x})(y_i - \bar{y})}{\sqrt{\sum (x_i - \bar{x})^2 \sum (y_i - \bar{y})^2}}, \tag{1}$$

where $x_i$ represents the expression of gene $x$ in cell $i$, and $\bar{x}$ represents the mean expression of gene $x$ in all cells. A supergene can be defined as a group of the $k$ most correlated original genes. As shwon in Fig 1B, typically the number of supergenes $l$ decreases with increasing $k$, and the rate of change gradually diminishes. Using sliding average algorithm, when the difference between the average $\bar{l}$ values of two consecutive windows, $\bar{l}_i$ and $\bar{l}_{i+1}$, is less than 2, the $\bar{k}_{i+1}$ and $\bar{l}_{i+1}$ values of the latter window are chosen as the optimal $k$ and $l$. For each supergene in a cell, the mean expression level of all genes within the class is considered as the supergene's

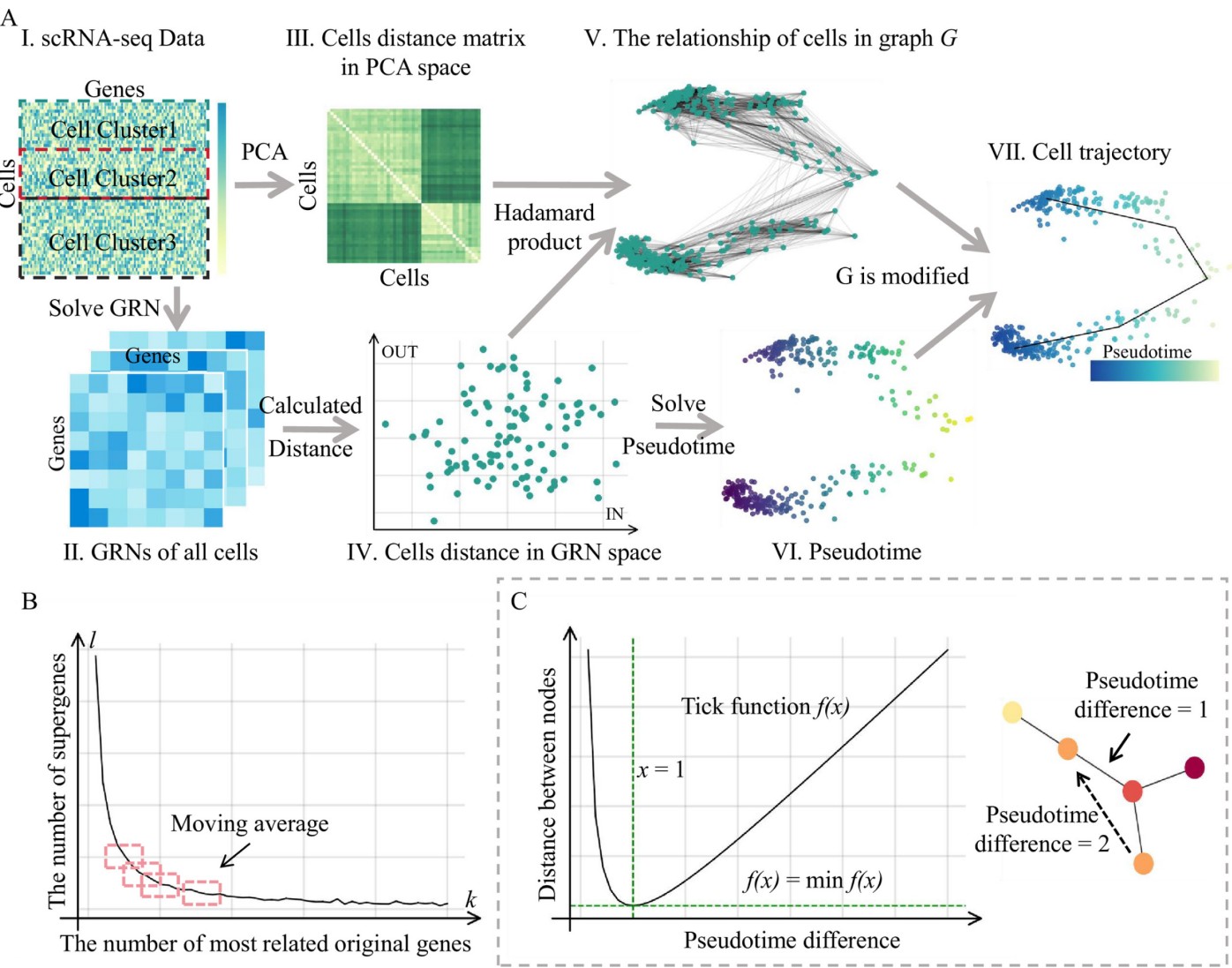

**Fig 1.** (A) Starting from single-cell data (I), we obtain the distance matrix of cells in PCA space (III) and the GRN of individual cells (II). By solving the Euclidean distance of cells in the space characterized by the out-degree and in-degree of all genes, we obtain the cell distance matrix (IV). The Hadamard product of the distance matrix in PCA space and the distance matrix in GRN space, with a constraint on the number of neighbors, results in the cell differentiation graph $G$ (V). Simultaneously, we solve the rough pseudotime of cells based on their relationships in GRN space (VI). We use the rough pseudotime to correct the cell differentiation graph $G$ and then find the minimum spanning tree of $G$ to obtain the cell differentiation trajectory graph (VII). (B) The curve of the number $k$ of supergenes obtained during gene pooling with respect to the number $l$ of original genes in each group. We select the optimal supergenes and the number of original genes per group by calculating the mean of supergenes in the preceding and following sliding windows. (C) The function graph of the tick function, which attains its minimum value at $x = 1$. We consider that when the pseudotime difference between clusters is 1 at the cluster scale, the distance between the two clusters is minimized, meaning the probability of a connecting edge between the two clusters is maximized.

expression level. Compared to other dimension reduction methods such as the PCA, gene pooling retains better interpretability while reducing dimensions.

To obtain the GRN in a single cell, we extend the method for solving GRNs from [33] to scenarios with only gene expression data. Based on the expressions of supergenes in each cell, we further construct a GRN for the cell using a ordinary differential equation

$$\frac{dx_i}{dt} = f_i(X_i) - \gamma x_i, \tag{2}$$

Here, we assume that the generation rate of gene $i$ is a result of its regulation by all other genes $X_i$ (the set of all genes excluding gene $i$), approximated as a function of the expression levels of these other genes, $f_i(X_i)$, the first-order Taylor expansion of which can be written as

$$f_i(X_i) = f_i(X_i^0) + \sum_{\substack{j=1 \\ j \neq i}}^{m} A_{ij}(x_j - x_j^0) + o(x_j - x_j^0), \tag{3}$$

where $m$ is the number of supergenes, $o(x_j - x_j^0)$ is the higher order infinitely small quantity of $x_j - x_j^0$, and $A_{ij}$ is the coefficient to be solved. Consider that in the steady state $\frac{dx_i}{dt} = 0$, then we can get

$$x_i = [f_i(X_i^0) - \sum_{\substack{j=1 \\ j \neq i}}^{m} A_{ij}x_j^0 + \sum_{\substack{j=1 \\ j \neq i}}^{m} A_{ij}x_j]\frac{1}{\gamma}, \tag{4}$$

since $x_j^0$ and $X_i^0$ are constants, $f_i(X_i^0) - \sum_{\substack{j=1 \\ j \neq i}}^{m} A_{ij}x_j^0$ can be denoted as $A_{i0}$. The degradation rate $\gamma$ is considered as constant. The $x_i$ can be further expressed as:

$$x_i = A_{i0} + \sum_{\substack{j=1 \\ j \neq i}}^{m} A_{ij}x_j. \tag{5}$$

Subject to the conservative condition $\sum_{\substack{j=1 \\ j \neq i}}^{m} A_{ij} = 0$, the coefficient matrix $A$ is calculated by minimizing the mean square error:

$$\{A_{ij}^0, A_{ij}\} = argmin\{[A_{i0} + \sum_{\substack{j=1 \\ j \neq i}}^{m} A_{ij}x_j - x_i + \lambda\sum_{\substack{j=1 \\ j \neq i}}^{m} A_{ij}]^2\}, \tag{6}$$

For the solved GRN (Fig 1A (II)), represented by the matrix $A = (A_{ij})$, we define: Intensity $o_i^k$ of the regulation of gene $i$ of cell $k$ on other genes and the intensity $i_i^k$ of the regulatory gene $i$ of other genes respectively, where $o_i^k = \sum_{i=1}^{m} |A_{ij}|$, and $i_i^k = \sum_{j=1}^{m} |A_{ij}|$. The regulatory intensity vectors and regulated intensity vectors of all genes in single cell $k$ are denoted as $I_k = (i_1^k, i_2^k, \cdots, i_m^k)$ and $O_k = (o_1^k, o_2^k, \cdots, o_m^k)$.

## Constructing the K-nearest neighbor graph of cells

In the linear space (GRN space) composed of $I$ and $O$, cell $k$ can be represented as $(I_k, O_k)$. We consider the binormal of $I$ and $O$ between cells to measure the distance $D^{GRN}$ between cells (Fig 1A (IV)), which reflects the dissimilarity of gene regulatory network between cells, that is, the dissimilarity of state changes between cells:

$$D_{kl}^{GRN} = \sqrt{(I_k - I_l)^2 + (O_k - O_l)^2}. \tag{7}$$

The distance between cells in the GRN space reflects to some extent the differences in gene regulatory relationships within cells. In addition, the static distance between cells in the expression space, $D^{PCA}$, can also be calculated by performing PCA on the gene expression matrix and then computing the Euclidean distance between cells in the PCA-transformed space (Fig 1A (III)).

Based on the similarity of cells at the current time and their similarity in future state transitions, the distance between cells $k$ and $l$ in the differentiation space can be obtained by

combining the above two measures, i.e., $D = D^{PCA} \cdot D^{GRN}$, where "·" represents the Hadamard product. Furthermore, based on the distances between cells in the differentiation space, the k-nearest neighbor graph of cells can be constructed (Fig 1A (V)), i.e., for each cell, edges are established with its $K$ nearest neighboring cells, where the edge weights correspond to the distances between them. Here, $k = 0.1n$, where $n$ is the number of cells.

## Calculation of pseudotime

Based on the similarity of state changes between cells $k$ and $l$, i.e., $D_{kl}^{GRN}$, we define the probability of cell $k$ transferring to another cell (e.g., $l$) as

$$P_{kl}^{GRN} = \frac{1/D_{kl}^{GRN}}{\sum_{l=1}^{n} 1/D_{kl}^{GRN}}. \tag{8}$$

Thus, the transfer entropy $S_k$ of the cell $k$ can be calculated according to the transition matrix,

$$S_k = -\sum_{l=1}^{m} P_{kl}^{GRN} log P_{kl}^{GRN}, \tag{9}$$

which is used to measure the uncertainty of disorder in the state transitions of cell $k$. Generally, cells with higher transfer entropy will transition to cells with lower entropy, e.g., stem cells typically have a more chaotic system state compared to terminal cells [34]. Due to the continuity of this process, the probability of transition and the difference in entropy between the two states should be a decreasing function. Thus, we construct the following function

$$P_{kl}^{G} = \begin{cases} e^{-P_{kl}^{GRN}(S_k - S_l)} & S_k > S_l, \{C_k, Cl\} \in G \\ 0 & S_k < S_l \end{cases} \tag{10}$$

to correct the transition probability from cell $k$ to cell $l$ in the KNN graph. We consider the infinite step transition probability of the cell in graph $G$,

$$M = \sum_{t=0}^{\infty} (P^G)^t = (1 - P^G)^{-1}. \tag{11}$$

For root cell $M_{(0)}$, we calculate rough pseudotime [21] $RPT_k = \|M_{(0)} - M_k\|_2$ for other cells, the rough pseudotime $RPT$ will be used as one of the features to construct the cell differentiation trajectory (Fig 1A (VI)).

## Trajectory inference

To simplify the computation process, we will calculate the cell differentiation trajectory at the cluster level. Therefore, we need prior knowledge of the cell clusters, which can either be pre-determined labels or obtained through clustering algorithms. For each cell cluster $Q$, we consider the connectivity of all nodes (cells) in the cluster, select the largest connected region in each cluster, and merge the cells in the connected region of a single cell cluster. The merging process follows the following rules. For the combination of nodes $C_{A_i}$ and $C_{A_j}$ in cell cluster $A$, node $C_{A_j}$ will be removed from graph $G$ and its edges will be spliced onto node $C_{A_i}$. For node $C_k$, if $\{C_k, C_{A_i}\} \in G$ and $\{C_k, C_{A_j}\} \in G$, the old edge between $C_{A_j}$ and $C_k$ is deleted and the weight of the new edge is

$$W'_{A_i k} = \frac{W_{A_i k} + W_{A_j k}}{4}. \tag{12}$$

If $C_k$ exists edge with only $C_{A_i}$ or $C_{A_j}$, then

$$W'_{A_i k} = \begin{cases} W_{A_i k} & \{C_k, C_{A_i}\} \in G \\ W_{A_j k} & \{C_k, C_{A_j}\} \in G \end{cases}. \tag{13}$$

Here we consider that the nodes (cells) with more edges to cluster $A$ should be closer to the merged cluster $A$. When all the nodes (cells) of the cluster are merged, we get a new relationship graph $G'$ between the clusters, the edges between the nodes (clusters) are derived from the merged result, and here we modify the edges according to the rough pseudotime. The pseudotime of the cluster is the mean of the rough pseudotime of all cells in the cluster. Here we use tick function $f(x) = x + \frac{1}{x}$, where $x$ represents the difference in pseudotime between clusters and $f(x)$ represents the distance between clusters (Fig 1C). Considering that $f(x)$ takes the minimum value at $x = 1$, that is, the distance between the two clusters calculated according to the pseudotime distance is the minimum, this is consistent with the continuity of the state transition of the cell. We normalized the calculated cluster pseudotime to the range of 0–10, then the distance $D^{PT}$ of the pseudotime difference between clusters can be obtained, and the final adjacency matrix of graph $G'$ between cell clusters can be obtained,

$$W = W^{G'} + 0.2 D^{PT}, \tag{14}$$

and then the transfer relationship between cell clusters can be obtained by calculating the minimum spanning tree for W (Fig 1A (VII)), that is, the differentiation trajectory graph $G^{MST}$ on the scale of cell clusters. According to $G^{MST}$, we delete the wrong edges on graph $G$ to obtain graph $G''$, and run the minimum spanning tree algorithm on graph $G''$ to obtain the differentiation trajectory on a single cell scale.

## Visualization of cell differentiation trajectories

By following the above steps, we can obtain the differentiation relationships between cell clusters. We utilize the differentiation relationships between clusters to modify the edges of the KNN graph at the single-cell level. We do not change the edges between cells within the same cluster, as cell differentiation is a continuous process and such transitions between cell states in different clusters are reasonable. If there is an edge between two cells but no edge between their corresponding clusters, we will remove the edge between the cells. This way, we can obtain the cell differentiation relationships at the single-cell level.

Based on the eigen decomposition of the matrix, we select the eigenvectors corresponding to the two largest eigenvalues other than 1. We then combine these with the results of UMAP dimensionality reduction to perform a two-dimensional visualization of the cell relationships. The method of combination involves normalizing the positional information corresponding to the eigenvectors and the positional information from UMAP to the same scale, for example, between -1 and 1, and then summing them to obtain the coordinate information for visualizing the cell positions.

The depiction of the differentiation trajectory relies on the information of cell clusters. We select the center of each cell cluster as the point through which the trajectory polyline needs to pass, and we draw the differentiation trajectory polyline based on the obtained differentiation relationships between cell clusters.

## Datasets and evaluation metrics

To assess the performance of scGRN-Entropy in inferring cell differentiation trajectories, we selected 14 real datasets from [14], where Saelens et al. developed a benchmark consisting of 110 real and 229 synthetic datasets for trajectory inference methods. Since the 110 real datasets include 8 different types of cell differentiation trajectories, we randomly selected a proportional number of datasets for each trajectory type (S1 Fig). Additionally, The datasets we used include four species (i.e. fly, human, macaque, and mouse) and covered six different sequencing technologies (S1 Table). Each dataset includes both gene expression counts for each cell and the cellular groupings, along with the connections (directed edges) between these groups (e.g., group 1 → group 2 → group 3). These directed edges serve as the ground truth. Accordingly, accuracy, defined as the ratio of correctly predicted edges to the total number of edges, was used as a metric to evaluate predictive performance.

# Results

## The overview of scGRN-entropy

scGRN-Entropy is a method designed for inferring cell differentiation trajectories and calculating pseudotime from scRNA-seq data, utilizing cell type labels and root cell information. Initially, we apply two dimensionality reduction techniques to the scRNA-seq data. The first method involves gene pooling, where resulting components, termed supergenes, retain biological interpretability while reducing dimensionality and grouping co-expressed genes. The second method is PCA, which determines static relationships between cells based on their gene expression values.

Following gene pooling, we construct a GRN composed of supergenes for each cell. Within the GRN space, we compute cell similarity metrics based on out-degree and in-degree, reflecting dynamic cell similarities. Integrating this dynamic similarity matrix with diffusion pseudotime allows us to infer the pseudotime of cell differentiation.

Additionally, combining dynamic similarity with a static similarity matrix yields a K-Nearest Neighbor (KNN) graph that describes cell differentiation. By merging nodes based on cell cluster information and edge attributes, we derive differentiation relationships between cells at the cluster level.

Finally, we visualize differentiation trajectories and pseudotime results through matrix decomposition and UMAP dimensionality reduction.

## Tested on real datasets

We verify the correctness and effectiveness of our method on multiple real datasets [14], which contain 8 types of trajectory (S1 Table). For the acyclic_graph trajectory type dataset, i.e. *neonatal-inner-ear-all_burns*, our method can establish most of the differentiation relationships between cell clusters (S2 Fig). Although it cannot fully construct the acyclic differentiation trajectory. For the two bifurcation trajectory datasets, i.e. *cellbench-SC1_luyitian* and *distal-lung-epithelium_treutlein* (S3 Fig), in the *cellbench-SC1_luyitian* dataset (Fig 2A and S3A Fig), the trajectory lines based on the centers of cell clusters effectively illustrate the differentiation pathways except the pseudotime coloring is somewhat chaotic. Similarly, in the *distal-lung-epithelium_treutlein* dataset (Fig 2B and S3B Fig), the pseudotime coloring is more distinct between different clusters. For the convergence trajectory type dataset, i.e. *olfactory-projection-neurons-DC3_VA1d_horns* (S4 Fig), which has two different starting points, DC3_24hAPF and VA1d_24APF, and a common endpoint, adPN adult, our method's main error is establishing differentiation relationships between the highly similar DC3_72hAPF and VA1d_72hAPF

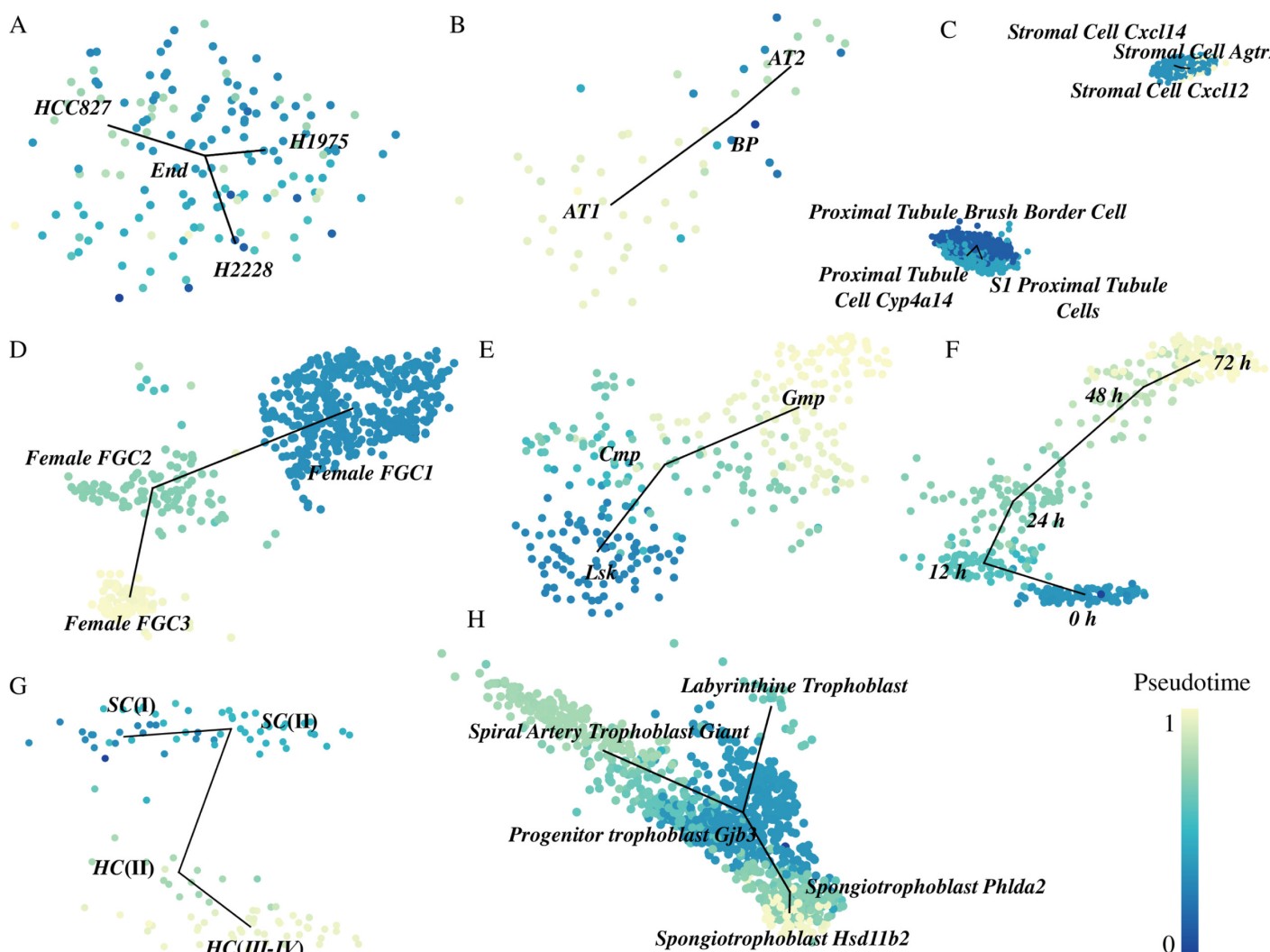

**Fig 2.** Differentiation trajectory plots for several datasets: (A) *cellbench-SC1_luyitian*, (B) *distal-lung-epithelium_treutlein*, (C) *mouse-cell-atlas-combination-5*, (D) *germline-human-female_li*, (E) *hematopoiesis-gates_olsson*, (F) *mESC-differentiation_hayashi*, (G) *neonatal-inner-ear-SC-HC_burns*, and (H) *placenta-trophoblast-differentiation_mca*.

clusters. For the cycle trajectory dataset, i.e. *cell-cycle_buettner* (S5 Fig), our method fails to identify the differentiation relationship between G2M and S2, which we attribute to the reliance on the minimum spanning tree algorithm. For the disconnected graph trajectory type datasets, i.e. *mouse-cell-atlas-combination-1* (S6A Fig) and *mouse-cell-atlas-combination-5* (Fig 2C and S6B Fig), our method can accurately infer the differentiation trajectories. However, the visualization does not perfectly distinguish between the internal cell clusters of the two disconnected parts. We attribute this to the limitations of UMAP and eigendecomposition methods in handling bipartite graphs. For the linear trajectory type datasets, i.e. *germline-human-female_li*, *hematopoiesis-gates_olsson*, *mESC-differentiation_hayashi*, and *neonatal-inner-ear-SC-HC_burns* (Fig 2D–2G and S7 Fig), our method successfully infers the differentiation trajectories, maintains good trajectory continuity and appropriate distances between cell clusters. For the *germline-human-female_li* dataset (Fig 2D and S7A Fig), we reconstructed the developmental process of human female fetal germ cells (FGCs). The individual embryo contains

several subpopulations simultaneously, highlighting the asynchrony and heterogeneity in the development of FGCs. In the *mESC-differentiation_hayashi* dataset (Fig 2F and S7C Fig), we replicated the developmental trajectory of mouse embryonic stem cells from 0 hours to 72 hours. We accurately identified cells at different time points and constructed a continuous cell developmental trajectory. Moreover, For the multifurcation trajectory datasets *placenta-trophoblast-differentiation_mca* (Fig 2H and S8 Fig), we accurately identified three branch lineages, including the complete differentiation process from Progenitor trophoblast Gjb3 to Spongiotrophoblast Phlda2 and then to Spongiotrophoblast Hsd11b2. For the tree trajectory type datasets, i.e. *epiblast-monkey_nakamura* and *hematopoiesis-clusters_olsson* (S9 Fig), due to the complexity of classifying cell clusters, it is hard to perfectly infer the cell trajectories at the cluster level.

## Comparison with existing methods

We compared our method with Slingshot [35], Monocle3 [36], and DBCTI [31] on these 14 datasets (S1 Table). For the 14 datasets, all three methods were run using default script parameters. We used accuracy, defined as the true positive rate of the inferred trajectories relative to the ground truth, to evaluate the performance of all methods (Table 1). The average prediction accuracy of scGRN-Entropy is 84.1%, significantly outperforming Slingshot (78.4%), Monocle3 (58.9%), and DBCTI (48.8%). It should be noted that for 7 of the datasets, Slingshot failed to produce valid inference results, with the R script returning the error "the system computation is singular". These cases were marked as "None," and the average accuracy was calculated based on the other 7 datasets where valid predictions were returned. Since scGRN-Entropy, Slingshot, and Monocle3 rely on the minimum spanning tree algorithm for trajectory inference, they cannot accurately predict cycle-type trajectories (e.g., for acyclic graph and cycle types), although they can infer all connections except for those connected to the closed cell groups, such as the cell transition between the G2M and S2 phases in the *cell-cycle_buettner* dataset (S5 Fig). Accurately predicting complex tree-type trajectories remains challenging for existing methods. For example, in the epiblast-monkey nakamura and hematopoiesis-clusters olsson datasets, the prediction accuracy of scGRN-Entropy, Monocle3, and DBCTI is below 50%. Although scGRN-Entropy achieves an accuracy of 43% on the epiblast-monkey

**Table 1. Comparison of scGRN-Entropy with existing state-of-the-art methods in terms of accuracy across 14 real datasets encompassing 8 types of differentiation trajectories.**

| Dataset | Trajector Type | scGRN-Entropy | Slingshot | Monocle3 | DBCTI |
|---|---|---|---|---|---|
| *neonatal-inner-ear-all_burns* [37, 38] | acyclic_graph | 67% | None | 67% | 67% |
| *cellbench-SC1_luyitian* [39, 40] | bifurcation | 100% | None | 33% | 100% |
| *distal-lung-epithelium_treutlein* [41] | bifurcation | 100% | None | 50% | 0% |
| *olfactory-projection-neurons-DC3_VA1d_horns* [42] | convergence | 63% | None | 25% | 38% |
| *cell-cycle_buettner* [43] | cycle | 67% | 67% | 67% | 100% |
| *mouse-cell-atlas-combination-1* [44] | disconnected graph | 100% | 57% | 41% | 21% |
| *mouse-cell-atlas-combination-5* [44] | disconnected graph | 100% | 25% | 67% | 27% |
| *germline-human-female_li* [45] | linear | 100% | 100% | 100% | 100% |
| *hematopoiesis-gates_olsson* [46] | linear | 100% | 100% | 100% | 67% |
| *mESC-differentiation_hayashi* [47] | linear | 100% | 100% | 75% | 44% |
| *neonatal-inner-ear-SC-HC_burns* [48, 49] | linear | 100% | None | 75% | 33% |
| *placenta-trophoblast-differentiation_mca* [44] | multifurcation | 100% | 100% | 67% | 50% |
| *epiblast-monkey_nakamura* [50, 51] | tree | 43% | None | 14% | 14% |
| *hematopoiesis-clusters_olsson* [52] | tree | 38% | None | 44% | 22% |

nakamura dataset, significantly higher than the 14% accuracy of the other two methods (Table 1), the inferred trajectory still deviates substantially from the experimental results; see S9 Fig. S2–S9 Figs in the Supplementary Materials also show the comparisons between the trajectories inferred by scGRN-Entropy and Monocle3 and the ground truth across the 14 datasets, suggesting that scGRN-Entropy provides more accurate predictions of cell differentiation trajectories, especially for linear types.

To demonstrate the importance of incorporating dynamic cell relationships, we also tested the scGRN-Entropy method without considering the dynamic cell relationships obtained from the GRN space (i.e., only using the cell relationship matrix in PCA space, treated as a baseline) on both linear and non-linear trajectory types. As shown in S10 Fig in the Supplementary Materials, for *mESC-differentiation_hayashi* (linear type), the baseline model predicted an evolution direction opposite to the experiments at 48h and 72h. For *distal-lung-epithelium_treutlein* (bifurcation type) dataset, the baseline predictions incorrectly inferred the direction of terminal differentiation path, suggesting that BP first differentiates into AT1 and then into AT2. For the multifurcation-type dataset splacenta-*trophoblast-differentiation_mca*, the predicted progenitor trophoblast Gjb3 cluster directly differentiated into the spongiotrophoblast hsd11b2 cluster, bypassing the spongiotrophoblast phlda2 cluster. In contrast, for all the three datasets, the scGRN-Entropy method accurately predicted the differentiation trajectories, fully aligning with experimental results (i.e., 100% accuracy; see Table 1). Moreover, compared to the baseline results, scGRN-Entropy better distinguished the sequential order of differentiation among individual cells in terms of pseudotime (S10 Fig). This demonstrates that the inclusion of GRN information significantly improved prediction accuracy across both linear and non-linear trajectory types, suggesting that GRNs may serve as an intrinsic driving force of cell differentiation. We solve for the transition probability matrix, transfer entropy, and cell pseudotime through the $D^{PCA}$ matrix, construct the k-nearest neighbors (KNN) graph for the cells, and obtain the differentiation trajectory using the minimum spanning tree algorithm. It can be observed that for the hematopoiesis-gates olsson dataset, the baseline method does not infer the cell differentiation trajectory well, and the pseudotime values show no distinction, meaning that the differentiation relationships between cells cannot be discovered (S10 Fig).

## Case study

We analyzed the *germline-human-female li* dataset [45], selecting the top 20 most and least correlated genes with pseudotime based on gene expression correlations (Fig 3A). We observed that during the transitional period, the expression of the selected genes in the Female FGC2 cell cluster was highly disordered. Some genes only exhibited high expression at the early and late stages of differentiation. Based on the analysis of the in-degree and out-degree of genes in the gene regulatory network, we found that the pooled supergenes, similar to specifically expressed genes, exhibited significant expression differences across different differentiation stages (Fig 3B).

Furthermore, we performed enrichment analysis on this dataset to explore how accurate trajectory inference contributes to understanding the roles of these genes in cellular signal transduction, metabolism, and other complex biological processes. First, we calculated the correlation between each gene's expression level across all cells and the inferred pseudotime. Using a significance threshold of 0.05, we selected genes that showed a significant positive or negative correlation with pseudotime. Enrichment analysis was then conducted on these two gene sets separately using EnrichR [53, 54] (Fig 4 and S2–S4 Tables). Among the genes positively correlated with pseudotime, KEGG analysis [55–59] revealed pathways related to the development of female reproductive organs, such as Progesterone-mediated oocyte

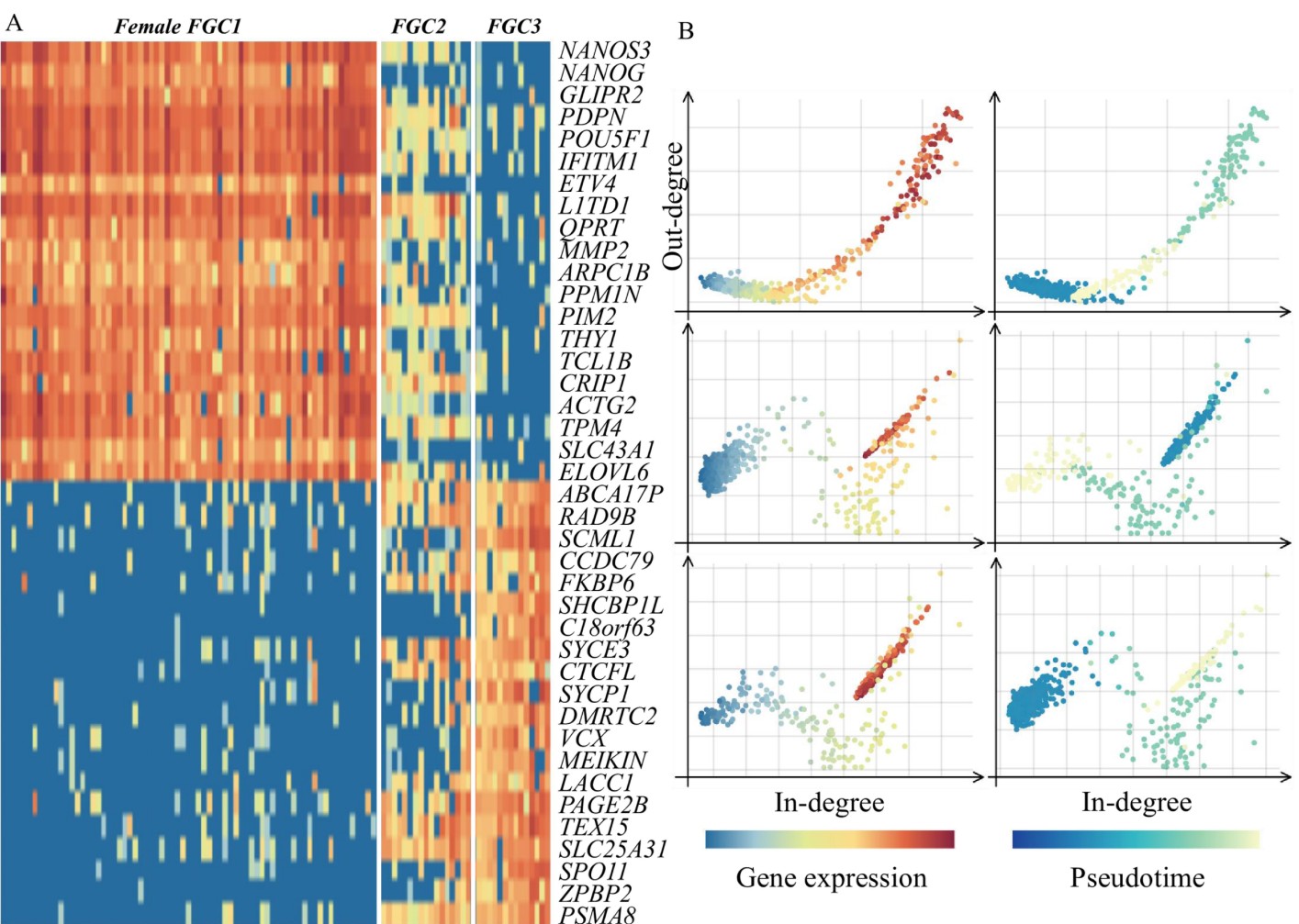

**Fig 3.** (A) The expression patterns of the 20 genes most positively correlated with pseudotime and the 20 least correlated genes clearly reflect the characteristics of cells at different stages: Female FGC1, FGC2, and FGC3. Genes highly expressed in the early stage (Female FGC1) gradually decrease in expression over time, whereas genes highly expressed in the later stage (Female FGC3) show a gradual increase. During the transitional stage (Female FGC2), all 40 genes display varying levels of expression. (B) The in-degree and out-degree plots for supergenes, shown on the horizontal and vertical axes, respectively, reveal that supergenes highly expressed during the transition period exhibit an increase in in-degree followed by a gradual decline. In contrast, supergenes highly expressed in the early and late stages demonstrate opposite trends in both in-degree and out-degree.

maturation (Homo sapiens hsa04914), Oocyte meiosis (Homo sapiens hsa04114), Cell cycle (Homo sapiens hsa04110), and FoxO signaling pathway (Homo sapiens hsa04068) (Fig 4A). As shown in Fig 4B, GO analysis of the positively correlated genes identified biological processes related to gonadal development and the cell cycle (with p-values/adjusted p-values < 0.001). These processes include homologous chromosome pairing at meiosis (GO:0007129), piRNA metabolic process (GO:0034587), homologous chromosome segregation (GO:0045143), meiosis I (GO:0007127), meiotic DNA double-strand break formation (GO:0042138), meiotic sister chromatid cohesion (GO:0051177), and ncRNA metabolic process (GO:0034660); see S3 Table in the Supplementary Materials. In the top 10 pathways with the smallest p-values, there are many molecular function pathways related to cell growth and development. For genes negatively correlated with pseudotime, KEGG analysis only identified the Focal adhesion (Homo sapiens hsa04510) pathway as being related to reproductive

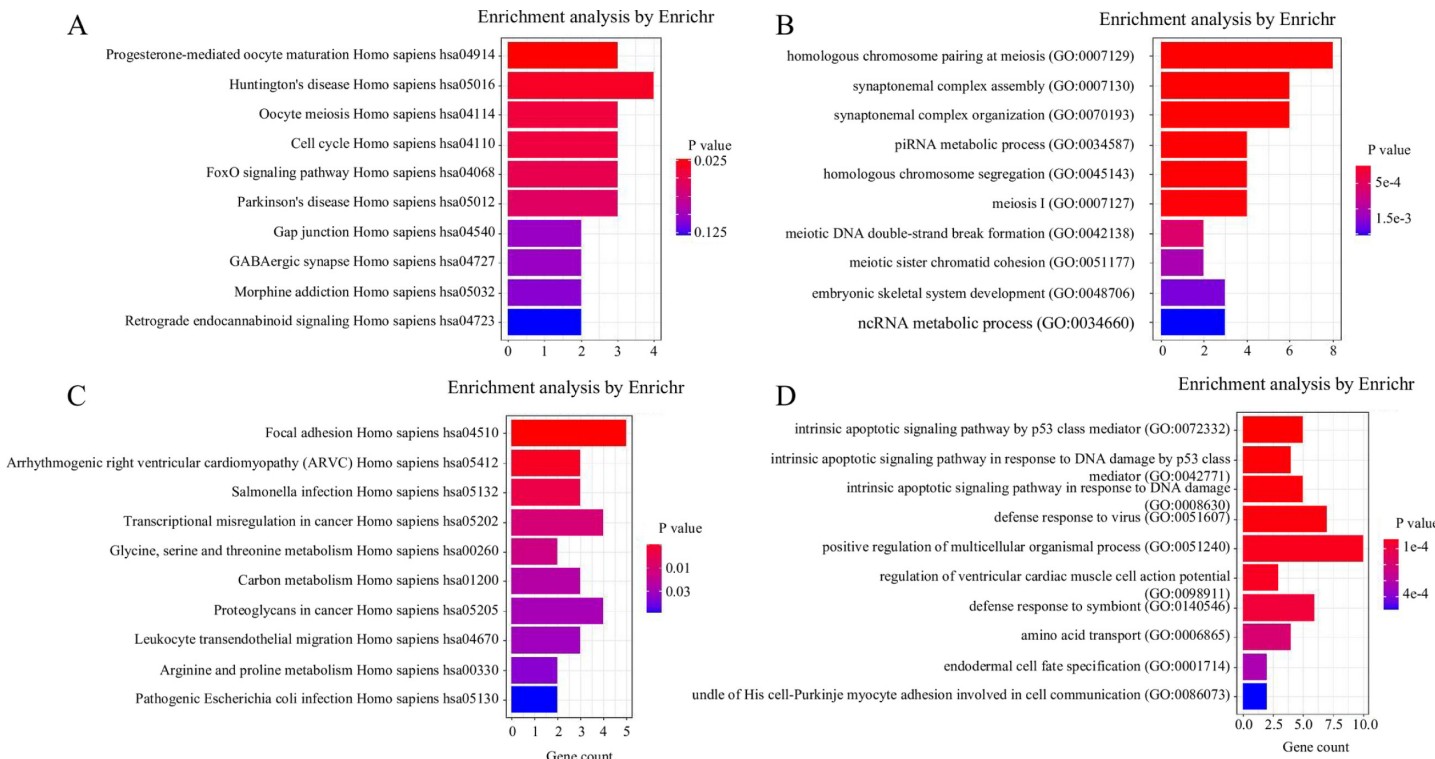

**Fig 4.** (A) KEGG enrichment analysis results of genes that were positively correlated with pseudotime, using the enrichment dataset "KEGG 2016". (B) GO enrichment analysis results of genes that were positively correlated with pseudotime, using the enrichment dataset "GO Biological Process 2021". (C) KEGG enrichment analysis results of genes that were negatively correlated with pseudotime, using the enrichment dataset "KEGG 2016". (D) GO enrichment analysis results of genes that were negatively correlated with pseudotime, using the enrichment dataset "GO Biological Process 2021".

development, with no statistically significant pathways related to the cell cycle (Fig 4C). GO analysis confirmed significant associations with apoptotic processes, such as the intrinsic apoptotic signaling pathway mediated by p53 (GO:0072332) and the defense response to viruses (GO:0051607); see Fig 4D and S4 Table in the Supplementary Materials. Unfortunately, KEGG analysis for both positively and negatively correlated genes did not get small enough p-value, which may be due to the relatively small number of genes selected ($< 100$). The results shown in the figure were exported from R, and the specific data can be found in S2–S5 Tables.

## Validation of GRN inference methods on simulated data

We used the Gillespie algorithm to simulate the expression control of five genes in the network to simulate the up-regulation and down-regulation of different gene expressions in the process of cell differentiation (S11 Fig). The stoichiometric matrix and propensity function of the chemical reaction (Table 2) are as follows [22]. Where $x_i$ represents the expression of *Genei*. The initial values we selected for $x_1$, $x_2$ and $x_3$ are sampled from uniform distributed between 40 and 80, and the initial values for $x_4$ and $x_5$ are 0. Based on our solution of the GRN, we found that over time, the gene regulation within the cell gradually stabilizes. Additionally, the changes in gene regulation are very drastic in the initial stage (Fig 5), which aligns with the fact that the cell's state is relatively chaotic during the early stages of differentiation.

**Table 2. Chemical reactions and their propensity functions.**

| Reaction | $x_1$ | $x_2$ | $x_3$ | $x_4$ | $x_5$ | Propensity Functions |
|---|---|---|---|---|---|---|
| Reaction1 | -1 | 0 | 0 | 0 | 0 | $0.005 \times x_1$ |
| Reaction2 | 0 | -1 | 0 | 0 | 0 | $0.005 \times x_2$ |
| Reaction3 | 0 | 0 | -1 | 0 | 0 | $0.005 \times x_3$ |
| Reaction4 | 0 | 0 | 0 | -1 | 0 | $0.01 \times x_4$ |
| Reaction5 | 0 | 0 | 0 | 0 | -1 | $0.01 \times x_5$ |
| Reaction6 | 1 | 0 | 0 | 0 | 0 | $1 - hill(0.1 \times x_3, 2) + hill(0.025 \times x_4, 2) - hill(0.025 \times x_5, 4)$ |
| Reaction7 | 0 | 1 | 0 | 0 | 0 | $1 - hill(0.1 \times x_1, 2) + hill(0.025 \times x_5, 4)$ |
| Reaction8 | 0 | 0 | 1 | 0 | 0 | $1 - hill(0.1 \times x_2, 2) - hill(0.025 \times x_4, 2)$ |
| Reaction9 | 0 | 0 | 0 | 1 | 0 | $hill(0.013 \times x_1, 8) + hill(0.025 \times x_4, 2)$ |
| Reaction10 | 0 | 0 | 0 | 0 | 1 | $hill(0.013 \times x_2, 2) + hill(0.025 \times x_5, 4)$ |

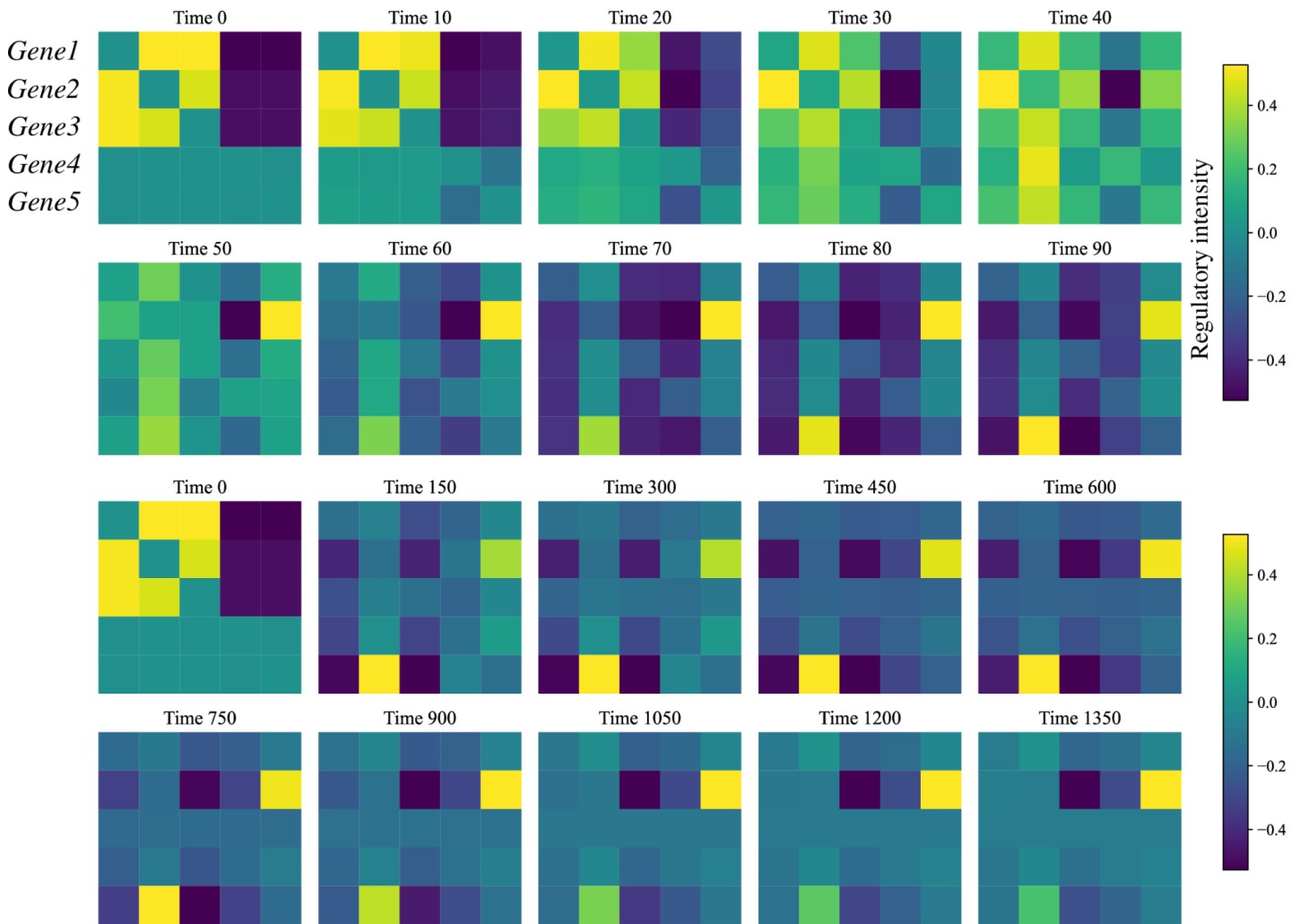

**Fig 5. Evolution of gene-gene regulatory strength over time, showing that the gene regulatory network underwent significant changes initially.** After a certain period, only Gene2 continued to exhibit slight regulatory influence on other genes until it eventually stabilized.

## Discussion and conclusion

In this paper, we proposed a new trajectory inference method for scRNA-seq data. The method distinguishes itself from existing methods by integrating relationships within both the GRN space and gene expression space to elucidate cell differentiation relationships. Using the cell entropy derived from the cell transition probability matrix, we constrain the state transition dynamics of cells to determine pseudotime. During the correction of the trajectory used pseudotime, we employed the tick function, which appropriately captured the relationship between pseudotime and trajectory at the cluster level. Finally, we derived the differentiation relationships at the single-cell level based on the differentiation relationships at the cluster level.

We validated the effectiveness of our method on 14 real-world datasets covering various trajectory types, successfully obtaining continuous cell differentiation trajectories and pseudotime. Although scGRN-Entropy has certain limitations when inferring circular and complex tree-type trajectories (e.g., it cannot form closed trajectories), it achieved an average prediction accuracy of 84.1%, which is significantly higher than that of current state-of-the-art methods. Additionally, our inference method relied on solving the GRN, involving matrix inversion during optimization, so the consumption of time and computational resources will be positively correlated with the number of target genes being resolved.

The dynamic changes in gene expression during cell differentiation are crucial for understanding cellular and organismal life processes [60, 61]. Our work not only introduces a new trajectory inference method but also highlights the contribution of dynamic changes in the GRN over time to cell differentiation. Comparisons with the baseline results demonstrate that the inclusion of GRN information significantly enhances the accuracy of scGRN-Entropy in predicting differentiation trajectories and provides a more detailed depiction of the order of cell differentiation. This suggests that dynamic gene regulation may be one of the primary driving forces of cell differentiation. Unfortunately, the current method is still limited in its ability to fully resolve the complex regulatory networks involving thousands of genes that are simultaneously expressed in a cell at any given moment [62, 63]. Nevertheless, scGRN-Entropy reduces the dimensionality based on gene expression relationships, aiming to retain the interpretability of the principal components while also preserving as much of the gene regulatory network's integrity as possible. In summary, the stochastic expression of genes determines cell fate. Understanding cell fate through gene relationships, and then analyzing gene relationships based on cell fate, holds significant importance for more detailed future studies of life processes.

## Supporting information

**S1 Fig. The proportion of the eight types of cell differentiation trajectories in the datasets proposed by Saelens et al. and the selected subsets used in our study.** Since the acyclic_-graph, convergence, and cycle trajectory types have very little data (the numbers are 1, 1, 2, respectively), our pro-portion on these trajectory types is significantly higher than the proportion of the original 110 real data sets.
(TIF)

**S2 Fig. Comparison of the differentiation trajectories inferred by scGRN-Entropy (middle) and Monocle3 (right) with the ground truth trajectory (left) for the *neonatal-inner-ear-all_burns* dataset (acyclic_graph type).**
(TIF)

**S3 Fig. Comparisons of the differentiation trajectories inferred by scGRN-Entropy (middle) and Monocle3 (right) with the ground truth trajectory (left) for two bifurcation type**

datasets: (A) the *cell-bench-SC1_luyitian* dataset, and (B) the *distal-lung-epithelium_treutlein* dataset.
(TIF)

**S4 Fig. Comparison of the differentiation trajectories inferred by scGRN-Entropy (middle) and Monocle3 (right) with the ground truth trajectory (left) for the *olfacto-ry-projection-neurons-DC3_VA1d_horns* dataset (convergence).**
(TIF)

**S5 Fig. Comparison of the differentiation trajectories inferred by scGRN-Entropy (middle) and Monocle3 (right) with the ground truth trajectory (left) for the *cell-cycle_buettner* dataset (cycle).**
(TIF)

**S6 Fig. Comparisons of the differentiation trajectories inferred by scGRN-Entropy (middle) and Monocle3 (right) with the ground truth trajectory (left) for two disconnected graph type datasets: (A) the *mouse-cell-atlas-combination-1* dataset, and (B) the *mouse-cell-atlas-combination-5* dataset.**
(TIF)

**S7 Fig. Comparisons of the differentiation trajectories inferred by scGRN-Entropy (middle) and Monocle3 (right) with the ground truth trajectory (left) for 4 linear type datasets: (A) the *germline-human-female_li* dataset, and (B) the *hematopoiesis-gates_olsson* dataset. (C) the *mESC-differentiation_hayashi* dataset, and (D) the *neonatal-inner-ear-SC-HC_burns* dataset.**
(TIF)

**S8 Fig. Comparison of the differentiation trajectories inferred by scGRN-Entropy (middle) and Monocle3 (right) with the ground truth trajectory (left) for the *placen-ta-trophoblast-differentiation_mca* dataset (multifurcation type).**
(TIF)

**S9 Fig. Comparisons of the differentiation trajectories inferred by scGRN-Entropy (middle) and Monocle3 (right) with the ground truth trajectory (left) for two tree type datasets: (A) the *epi-blast-monkey_nakamura* dataset, and (B) the *hematopoiesis-clusters_olsson* dataset.**
(TIF)

**S10 Fig.** **(A) For the *mESC-differentiation_hayashi* dataset, we obtained differentiation trajectories without considering the impact of GRN on the transition probability matrix, distance matrix, and pseudotime. We then compared these with trajectory inferences and pseudotime calculations con-sidering GRN's influence. Without GRN, there were significant errors at 48h and 72h, where cells at 24h directly transitioned to 72h and then to 48h. However, with GRN, a continuous differentiation trajectory was achieved. (B) In the *dis-tal-lung-epithelium_treutlein* dataset, we examined differ-entiation trajectories without and with GRN's influence. BP cell clusters should differentiate into AT1 and AT2 clusters separately. Without GRN, BP first differentiated into AT1 and then into AT2. With GRN, pseudotime indicated a smaller time gap between AT1 and AT2 clusters. (C) For the *pla-centa-trophoblast-differentiation_mca* dataset, we analyzed differentiation trajectories without and with GRN's effects. Without GRN, the progenitor trophoblast Gjb3 cluster directly differenti-ated into the spongiotrophoblast hsd11b2 cluster, skipping the spongio-trophoblast phlda2 cluster. Pseudotime without GRN had only initial and final points,**

while scGRN-Entropy provided a more continuous pseudotime.
(TIF)

**S11 Fig. Based on our given stoichiometric matrix and reaction propensity functions, we obtained the time-series expression curves for 5 genes.** *Gene5* reached a stable state first, while *Gene2* reached a stable state only after all other genes had stabilized. Meanwhile, we assigned the gene expression values at each integer time point to individual cells.
(TIF)

**S1 Table. Summary of the datasets used in this study, including the number of cells, number of genes, types of differentiation trajectories, species, and sequencing technologies.**
(XLSX)

**S2 Table. Quantification of the top 10 pathways with the smallest p-values from KEGG analysis of the top 5% of genes that are positively correlated with pseudotime.**
(XLSX)

**S3 Table. Quantification of the top 10 pathways with the smallest p-values from GO analysis of the top 5% of genes that are positively correlated with pseudotime.**
(XLSX)

**S4 Table. Quantification of the top 10 pathways with the smallest p-values from KEGG analysis of the top 5% of genes that are negatively correlated with pseudotime.**
(XLSX)

**S5 Table. Quantification of the top 10 pathways with the smallest p-values from GO analysis of the top 5% of genes that are negatively correlated with pseudotime.**
(XLSX)

## Author Contributions

**Conceptualization:** Rui Sun, Bengong Zhang.

**Data curation:** Rui Sun, Wenjie Cao, ShengXuan Li.

**Formal analysis:** Rui Sun, Jian Jiang.

**Funding acquisition:** Bengong Zhang.

**Investigation:** Rui Sun, ShengXuan Li.

**Methodology:** Rui Sun, ShengXuan Li, Yazhou Shi.

**Project administration:** Jian Jiang, Yazhou Shi, Bengong Zhang.

**Resources:** Jian Jiang, Bengong Zhang.

**Software:** Rui Sun, Wenjie Cao, Jian Jiang.

**Supervision:** Bengong Zhang.

**Validation:** Jian Jiang, Yazhou Shi, Bengong Zhang.

**Visualization:** Rui Sun, ShengXuan Li.

**Writing – original draft:** Rui Sun, Bengong Zhang.

**Writing – review & editing:** Rui Sun, Yazhou Shi, Bengong Zhang.

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
