## [Decision Letter · Decision Letter 0]

16 Sep 2024

Dear Prof. Zhang,

Thank you very much for submitting your manuscript "scGRN-Entropy: Inferring Cell Differentiation Trajectories Using Single-Cell Data and Gene Regulation Network-Based Transfer Entropy" for consideration at PLOS Computational Biology.

As with all papers reviewed by the journal, your manuscript was reviewed by members of the editorial board and by several independent reviewers. In light of the reviews (below this email), we would like to invite the resubmission of a significantly-revised version that takes into account the reviewers' comments.

In particular, the reviewers raise concerns regarding benchmarking and validation of the approach. These issues should be addressed.

We cannot make any decision about publication until we have seen the revised manuscript and your response to the reviewers' comments. Your revised manuscript is also likely to be sent to reviewers for further evaluation.

Sincerely,

Saurabh Sinha

Academic Editor

PLOS Computational Biology

Stacey Finley

Section Editor

PLOS Computational Biology

Reviewer's Responses to Questions

**Comments to the Authors:**

Reviewer #1: Sun et al. introduce scGRN-Entropy, a cell trajectory inference method using scRNA-seq data. This method uses both static changes in gene expression and dynamic changes in gene regulatory networks to predict cell trajectory and pseudotime. scGRN-Entropy was validated using eight real benchmarking datasets that include linear and other non-linear trajectory types. The paper provides a detailed model description, and demonstrates its accuracy and biological relevance. However, a major concern lies in the selection of validation datasets, as the study could have leveraged more publicly available datasets, representing diverse organisms and trajectory types. Also, the figures are not presented in the order they are discussed, making the paper a bit difficult to follow.

[Major comments]

1. In Table 1, eight diverse datasets of different trajectory types were included, selected from the Saelens et al. 2019 benchmarking paper. In Saelens et al., 110 real datasets and 229 synthetic datasets used for comparisons were released online, along with the codes. Could the authors provide a rationale for the selection of these specific eight datasets? For example, why were tree-based trajectories excluded from the study? Are these datasets chosen because they represent typical scenarios in the biological systems of interest, or were other factors considered in the selection?

2. In Table 2, scGRN-Entropy was compared with other methods and showed accurate predictions on all 8 trajectories. In addition to trajectory, could the author also compare stability of predictions which is important for reproducibility in research? Also, adding a column specifying the trajectory type (bifurcation, linear, etc.) will enhance the interpretation of the results.

3. The uniqueness of scGRN-Entropy lies in its incorporation of cell distance in the GRN space. In Fig 2c, the author showed the decrease in performance when removing GRN information. Could the authors extend this analysis to other non-linear trajectory types? It will provide deeper insights into the effectiveness of GRN integration.

4. For the gene set enrichment analysis, are the p-values reported raw values or multiple testing corrected? Also, in the KEGG enrichment analysis, despite selecting genes from the top/bottom 0.05 quantiles, the reported pathways are enriched by fewer than five genes. The enrichR package offers many libraries of pathways, is there any other pathway that showed more significant enrichment?

5. Table 3 is not referenced anywhere in the manuscript. Also, both row and column labels are missing.

6. Making the codes of scGRN-Entropy publicly available would allow other researchers to validate the findings. The code and data availability link (https://zenodo.org/records/1443566%5C#.Y3q1fnbMKUl) isn’t working.

[Minor comments]

1. Fig 1B was introduced before Fig 1A.

2. In Methods- Datasets, it’s unclear why some datasets are categorized as gold while others as silver standard.

3. In the Fig 3 caption, the panel labels were not discussed in alphabetical order. This might be due to the shape and space constraints of the panels, but reordering them will improve clarity and readability.

4. How many supergenes were chosen in each of the validation dataset?

5. Fig 3F, the size of the dots is different from the other panels.

6. Fig 4B, the labels for FGC1, FGC2, FGC3 are missing.

7. Fig 2C was cited at the very beginning of the Results, while Fig 2A and 2B were introduced in the last subsection of Results.

Reviewer #2: The manuscript presents a novel approach, scGRN-Entropy, for inferring cell trajectories and pseudotime from scRNA-seq data. The incorporation of gene regulatory network (GRN) information is a valuable contribution to the field. While the methodology appears sound, additional validation and a more compelling application are necessary to strengthen the manuscript for publication.

1. The manuscript lacks a clear definition of ground truth for benchmarking. It is unclear how to determine true/false in this context. what means “completely correct” and “not completely correct”? Is it including miss ordering of even 1 cell considered as False? A quantitative metric is essential for objectively evaluating the performance of the proposed method compared to existing approaches.

2. Necessity of GRN Space Similarity. The authors should explore the impact of using only static similarity (without GRN space similarity) on the inferred pseudotime, KNN graph, and trajectories. This would help to elucidate the contribution of GRN information to the method's performance and identify potential limitations of relying solely on GRN-based similarity.

3. A more compelling application is needed. While the enrichment results obtained from the analysis in case study are interesting, it is important to demonstrate the unique advantages of the proposed method compared to existing approaches. The authors should discuss how their method can lead to novel discoveries or more reasonable results that are not achievable with current techniques.

**Have the authors made all data and (if applicable) computational code underlying the findings in their manuscript fully available?**

Reviewer #1: **No: **The code and data availability link (https://zenodo.org/records/1443566%5C#.Y3q1fnbMKUl) isn’t working.

Reviewer #2: **No: **The code is not provided.

PLOS authors have the option to publish the peer review history of their article (what does this mean?). If published, this will include your full peer review and any attached files.

Reviewer #1: No

Reviewer #2: **Yes: **Zhana Duren
---

## [Decision Letter · Decision Letter 1]

12 Nov 2024

Dear Prof. Zhang,

We are pleased to inform you that your manuscript 'scGRN-Entropy: Inferring Cell Differentiation Trajectories Using Single-Cell Data and Gene Regulation Network-Based Transfer Entropy' has been provisionally accepted for publication in PLOS Computational Biology.

Best regards,

Saurabh Sinha

Academic Editor

PLOS Computational Biology

Stacey Finley

Section Editor

PLOS Computational Biology

Feilim Mac Gabhann

Editor-in-Chief

PLOS Computational Biology

Jason Papin

Editor-in-Chief

PLOS Computational Biology

Reviewer's Responses to Questions

**Comments to the Authors:**

Reviewer #1: Thank the authors for addressing my previous concerns. I have no more comments.

Reviewer #2: All my comments are addressed. I have no further comments.

**Have the authors made all data and (if applicable) computational code underlying the findings in their manuscript fully available?**

Reviewer #1: Yes

Reviewer #2: Yes

PLOS authors have the option to publish the peer review history of their article (what does this mean?). If published, this will include your full peer review and any attached files.

Reviewer #1: No

Reviewer #2: No

---

## [Editor Report · Acceptance letter]

17 Nov 2024

PCOMPBIOL-D-24-01142R1 

scGRN-Entropy: Inferring Cell Differentiation Trajectories Using Single-Cell Data and Gene Regulation Network-Based Transfer Entropy

Dear Dr Zhang,

I am pleased to inform you that your manuscript has been formally accepted for publication in PLOS Computational Biology. Your manuscript is now with our production department and you will be notified of the publication date in due course.

With kind regards,

Zsofia Freund
